# Experiences of women in prenatal, childbirth, and postpartum care during the COVID-19 pandemic in selected cities in Brazil: The resignification of the experience of pregnancy and giving birth

Zeni Carvalho Lamy[1], Erika Barbara Abreu Fonseca Thomaz[1]*, Aluísio Gomes da Silva-Junior[2], Gisele Caldas Alexandre[2], Maria Teresa Seabra Soares de Britto e Alves[1], Ruth Helena de Souza Britto Ferreira de Carvalho[1], Letícia Oliveira de Menezes[3], Sandro Schreiber de Oliveira[4], Maurício Moraes[4], Yasmim Bezerra Magalhães[5], Tatiana Raquel Selbmann Coimbra[6], Lely Stella Guzman-Barrera[6]

1 Public Health Department, Universidade Federal do Maranhão, São Luís, MA, Brazil, 2 Universidade Federal Fluminense, Niterói, RJ, Brazil, 3 Universidade Católica de Pelotas, Pelotas, RS, Brazil, 4 Universidade Federal do Rio Grande, Pelotas, RS, Brazil, 5 Centro Universitário do Planalto Central Apparecido dos Santos, Uniceplac, Brasília, DF, Brazil, 6 Pan American Health Organization / World Health Organization (PAHO/WHO), Brasília, DF, Brazil

* erika.barbara@ufma.br

**Data Availability Statement:** Because this is a qualitative research, with in-depth interviews,

## Abstract

The COVID-19 pandemic has impacted public and private health systems around the world, impairing good practices in women's health care. However, little is known about the experiences, knowledge, and feelings of Brazilian women in this period. The objective was to analyze the experiences of women, seen at maternity hospitals accredited by the Brazilian Unified Health System (SUS, acronym in Portuguese), regarding health care during pregnancy, childbirth, and postpartum periods, their interpersonal relationships, and perceptions and feelings about the pandemic. This was a qualitative, exploratory research, carried out in three Brazilian municipalities with women hospitalized in 2020, during pregnancy, childbirth, or postpartum period, with COVID-19 or not. For data collection, semi-structured individual interviews (in person, by telephone, or by digital platform) were conducted, recorded and transcribed. The content analysis of thematic modalities was displayed as per the following axes: i) Knowledge about the disease; ii) Search for health care in prenatal, childbirth, and postpartum periods; iii) Experience of suffering from COVID-19; iv) Income and work; and v) Family dynamics and social support network. A total of 46 women were interviewed in São Luís-MA, Pelotas-RS, and Niterói-RJ. Use of media was important to convey information and fight fake news. The pandemic negatively impacted access to health care in the prenatal, childbirth, and postpartum periods, contributing to worsening of the population's social and economic vulnerabilities. Women experienced diverse manifestations of the disease, and psychic disorders were very frequent. Social isolation during the pandemic disrupted the support network of these women, who found social support strategies in communication technologies. Women-centered care–including qualified listening and mental health

addressing sensitive issues from the point of view of identifying women, and, considering that the statements that support our findings and conclusions were made available in the manuscript, we believe that the availability of the entire transcription of the interviews can violate the ethical precepts of guaranteeing the secrecy and privacy of the participants. So, we did not make available all the data (transcription of all speeches), but we have inserted several excerpts from the speeches of the women (de-identified) throughout the manuscript. The entire data can be accessed upon request through e-mail to a non-author contact: Sandra Santos, Pan-American Health Organization – PAHO, E-mail: sandra@paho.org, Phone: +55 61 32519513).

**Funding:** The authors are grateful to the Bill and Melinda Gates Foundation [INV-017424], World Health Organization (WHO), Pan American Health Organization (PAHO) [ZCL, EBAFT, AGSJ, GCA, MTSSBA, RHSBFC, LOM, SSO, MM, YBM, TRSC, LSGB], National Council for Scientific and Technological Development (CNPq, acronym in Portuguese) [processes 306592/2018-5 (EBAFT), 314939/2020-2 (ZCL), 311479/2020-2 (MRSSBA), and 308917/2021-9 (EBAFT)], the Coordination for the Improvement of Higher Education Personnel (CAPES, acronym in Portuguese) [finance code 001] (EBAFT, MTSSBA, RHSBFC, ZCL), and Foundation for the Support of Scientific and Technological Research and Development of Maranhão (FAPEMA) for supporting scientific publication. We would also like to thank the Universidade Federal do Maranhão, Universidade Federal Fluminense, Universidade Federal of Rio Grande, Universidade Católica de Pelotas, Municipal Health Departments of São Luís, Niterói e Pelotas, the State Health Departments of Maranhão, Rio de Janeiro and Rio Grande do Sul. The funders had no role in study design, data collection and analysis, decision to publish, or preparation of the manuscript.

**Competing interests:** The authors have declared that no competing interests exist.

support–can reduce the severity of COVID-19 cases in pregnant, parturient, and postpartum women. Sustainable employment and income maintenance policies are essential to mitigate social vulnerabilities and reduce risks for these women.

## Introduction

The COVID-19 pandemic has impacted public and private health systems around the world [1–4]. Faced with the large and unexpected demand for medium- and high-complexity services, the healthcare network and clinical protocols required reorganization to meet the growing number of cases of the disease, reduce the high mortality rates, and maintain the delivery of essential services, such as prenatal care and childbirth [5–12].

The World Health Organization (WHO) / Pan American Health Organization (PAHO) initiative "*COVID-19*: *Mitigating indirect effects on essential health services for neonates*, *children*, *adolescents and the elderly*" [13], funded by the Bill Gates Foundation, supported 20 priority countries. For the Region of the Americas, Bolivia and Brazil were selected. The initiative evaluated (and fostered) the maintenance of essential services as a result of the spread of the COVID-19 pandemic, identifying how the pandemic has affected achieving the goals defined in the Sustainable Development Goals. Therefore, it aimed to find and contextualize actions related to delivery of essential health services for vulnerable populations, and it identified the experiences in health care of women during pregnancy, childbirth, and postpartum periods, as well as of children as one of the priority areas.

In Brazil, evidence points to changes in care for women and children [14, 15], with potential damage to good care practices, beginning in the prenatal care [16]. At first, with all efforts aimed to fight the pandemic, there were bottlenecks in providing outpatient and hospital services considered elective, as well as a lack of an organized and structured network of primary services to ensure provision of health care. In this context, it was necessary to consider how Brazilian health services were organized to maintain care in response to situations of greater vulnerability, given the worsening of access and quality problems that already existed before the pandemic [17], including access barriers to mechanical ventilation and intensive care for obstetric patients [18]. Three situations were identified as key aspects responsible for maternal deaths in Brazil. First, delay in identifying and testing pregnant women with COVID-19 symptoms. Second, delay in hospital admission of women diagnosed with COVID-19. Third, delay in providing timely treatment at intensive care units (ICU) [19]. Therefore, appropriate management of pregnancy during the COVID-19 pandemic was a tough issue [20].

The challenge for the Brazilian Unified Health System (SUS) was how to effectively respond to a disease that, despite not having high case fatality rates, could cause the collapse of the health system. The geographic characteristics, population size, and the Brazilian regional peculiarities brought additional difficulties to tackle the problem, which should be dealt with according to the specificities of each location. Another challenge is the context of an overloaded health system with budgetary constraints. The search for effective strategic planning must consider the guiding principles of the SUS and the impact scenario, not only in the dimensions of morbidity and mortality, but also essentially in the context and dynamics of people's lives and their social and economic consequences.

Therefore, given the problems related to health care providers to women [14–16, 18, 20], the increased risk of dying [21, 22], the psychological suffering brought about by the pandemic [23–25], and the uncertainties experienced in this period [5], several studies have investigated

the experiences, knowledge, and feelings of these women during pregnancy, childbirth and postpartum periods [23, 25–34]. However, we did not identify other qualitative studies, using triangulation of methods and data collection techniques, with a multicentric approach, which had interviewed women at different levels of obstetric risk, living in Brazilian municipalities with different sociodemographic characteristics, with and without a history of infection by COVID-19 during pregnancy or hospitalization for childbirth. Hence, the objectives of this study were to analyze the experiences of women regarding health care provided to pregnant, parturient, and postpartum women, their interpersonal and family relationships, and the perceptions of risk and feelings related to the environment of coping with the pandemic, in three Brazilian municipalities. The study raised key issues for maternal and child care and provided subsidies for planning, monitoring, and evaluation of physical and mental health policies, aiming to mitigate problems, enhance quality of care, and coordinate income and employment maintenance policies.

## Methods

### Research type, period, and settings

This was a qualitative, exploratory research, carried out from November 2020 to May 2021, at SUS-accredited maternity hospitals in the municipalities of São Luís—Maranhão (MA), Pelotas—Rio Grande do Sul (RS) and Niterói—Rio de Janeiro (RJ). The study is part of the PAHO/WHO project entitled "*Identification of the indirect effects of COVID-19 on essential services for pregnant women, neonates, children, adolescents and the elderly at the subnational level in Brazil*", as part of the WHO initiative "*COVID-19: Mitigating indirect effects on essential health services for neonates, children, adolescents and the elderly*" [13].

In São Luís, with an estimated population of 1,115,932 inhabitants in 2021 [35], the study included two high-complexity maternity hospitals, referral centers for COVID-19 patients, which provided isolation areas for pregnant and postpartum women with suspected or confirmed COVID-19, as well as a Maternal and Neonatal Intensive Care Unit (ICU).

In Pelotas, with an estimated population of 343,826 inhabitants in 2021 [35], the study included two maternity hospitals that attend to patients through the SUS, but only one of them was used as a referral center for COVID-19 patients. However, patients who had COVID-19 at some point during pregnancy, childbirth, or postpartum period were included, regardless of the maternity hospital they were admitted to.

In Niterói, with an estimated population of 516,981 inhabitants [35], a low- and moderate-risk maternity hospital was included, which is a referral center for childbirth in the municipality. Although it was not the referral maternity hospital for care of women with COVID-19, isolated spaces were created for management of symptomatic patients, whenever necessary. During the study, there were no hospitalizations of women with COVID-19.

The diverse characteristics of the maternity hospitals expanded the scope of the study and analysis of different experiences by women.

### Study participants and sampling

The participants were women attended to at maternity hospitals of the study. Both women admitted for childbirth or treatment of COVID-19, during pregnancy, childbirth, and postpartum periods, from March to November 2020, were included.

The sample was intentionally defined considering the sociodemographic and clinical characteristics of the women, seeking to include different aspects, such as phases of the pregnancy-puerperal cycle, age, clinical status, marital status, number of children, education level, income, and place of residence, in addition to diagnosis of COVID-19. They were interviewed in three

municipalities; in that, 17 in São Luís, 12 in Pelotas, and 17 in Niterói. To close the sample, the meaning saturation technique was used, which indicates interrupting collection when further data do not bring new pieces of information to the object studied [36].

## Study techniques and tools

In the three municipalities of the research, an individual semi-structured interview was used, based on a previously prepared script with questions about the experiences of pregnancy, childbirth, and birth during the COVID-19 pandemic. Before the interviews, a structured questionnaire was filled out with sociodemographic, cultural, and clinical history data of each woman and neonate.

The interviews were conducted face-to-face, via telephone or digital platform, according to the interviewees' choice, during the most convenient days and times for them. For in-person interviews, all necessary safety measures (use of mask, distancing, hand washing, and use of alcohol-based hand sanitizer) were complied with. The average duration was 50 minutes, and the interviews were recorded and later transcribed. Each municipality had a team of duly trained interviewers.

## Data collection

The first step was to contact the research sites to identify the participants. This took place in the context of the development of the technical product "*Identification of strategies, experiences and evidence-based initiatives, which was already underway in the selected region, in Brazil*", to mitigate the indirect effects of the pandemic on five vulnerable population groups (pregnant women, newborns, children, adolescents and the elderly), as part of the response to the pandemic, according to the demographic, geographic, epidemiological and care contexts. In our research, a sub-project was created for women. A list of all women admitted to the maternity hospitals until the data collection starting date was made available. Based on this list, a review of medical records was conducted seeking patient data to define the sample. These data were organized in a table for each maternity hospital, categorized according to the characteristics found defining typologies, and women were separated into groups. Next, telephone contact was initiated to present the research and invitation to participate. There was no direct refusal. However, after three unsuccessful contact attempts, the participant was replaced by another from the same group of identity characteristics.

## Analysis

Content analysis was performed in the thematic modality as proposed by Bardin, 2011 [37], and Minayo, 2014 [38]. The results were presented through thematic axes, organized from the synthesis of the interviews.

## Ethical considerations

The research was approved by the Research Ethics Committee of the University Hospital of the Universidade Federal do Maranhão (CAAE 35645120.9.0000.5086, approved on August 25th, 2020, and amended on November 27th, 2020, to include Pelotas and Niterói). Besides, the research was further approved by the Research Ethics Committees of the Universidade Católica de Pelotas (CAAE 38281820.3.0000.5339, approved on November 9th, 2020) and the Pan American Health Organization (Ref. number PAHOERC.0260.02, on October 19th, 2020), in compliance with Resolutions 466/12 and 510/16 of the National Health Council. All study participants signed an informed consent form in two copies—one delivered to each participant,

and the other kept by the project team. A written assent was signed by parents or guardians in case of adolescent participants. Funders did not interfere in the methodology or any other step that could influence the results or conclusions of the study. The interviewees were granted anonymity. The names were replaced by codes that started with the first letter of the municipality (N–Niterói; P–Pelotas; and S–São Luís), followed by the serial number of the interview, age of the woman interviewed, and period the recorded text referred to (pregnancy, childbirth or postpartum). When the interviewee referred to the care received in the health care network, the setting of care (MRH—low- and moderate-risk maternity; MAR—high-risk maternity; and UBS—primary care unit, acronyms in Portuguese) was inserted at the end of the coding.

## Results and discussion

### Sociodemographic characteristics

We interviewed 46 women, aged 17 to 40 years, in the three municipalities, and the largest age group comprised those aged 24 to 34 years. The majority declared to be black or mixed race (in Pelotas, white), married or in a consensual union (living with a partner and children), Evangelical or Catholic, and residing mainly in urban areas (in São Luís, three women lived in rural areas). Most had completed high school, except in Pelotas, where a higher proportion of women had further education. Most said they were employed (including self-employed), with a monthly family income of up to three minimum wages and receiving emergency financial aid (in São Luís, incomes of up to one minimum wage predominated, and in Pelotas, there was a slight predominance of incomes of up to three minimum wages).

Most women were primiparous or had two children (in São Luís, one pregnant woman was multiparous with 16 pregnancies and 11 children). All had access to prenatal care but the number of visits varied from 4 to 10. Public primary care facilities were the main referral centers for prenatal care in all municipalities studied. In São Luís some pregnant women were also followed up in outpatient clinics of the maternity hospitals due to risk factors. In Pelotas and Niterói there was a greater number of visits to private facilities for prenatal care and ancillary tests.

Normal development of pregnancy was described by most women, with no complications and comorbidities, except for seven women in São Luís who reported comorbid conditions. In this same municipality, there was a higher proportion of cesarean deliveries and premature births (one stillbirth). Most deliveries in Pelotas and Niterói were vaginal and at term (in Pelotas, seven of the interviewees had not yet given birth at the time of the interview).

### Thematic axes derived from the interviewees' statements

The results are displayed by five thematic axes: i) Knowledge about the disease; ii) Search for health care in prenatal, childbirth and postpartum periods; iii) Experience of suffering from COVID-19; iv) Income and work; and v) Family dynamics and social support network (Fig 1).

**Knowledge about the disease, prevention, and risk perception.** Initially, the seriousness of the problem and news of deaths seemed far away to these women. News about severity of the pandemic came from abroad and its progression in Brazil was perceived with less impact. The fact that pregnant women had access to news about COVID-19 only through television and social media confirmed the idea that the disease was not yet part of their reality.

The proximity to the real risk of a little-known virus with unpredictable effects on the health of the population was realized when the pandemic was no longer only in the media, and reached the cities of residence of these women; that is, it began to affect well-known characters and to be witnessed in their own lives, and in the lives of relatives and close friends. Associated with this, local authorities began to impose measures to prevent the spread of the disease,

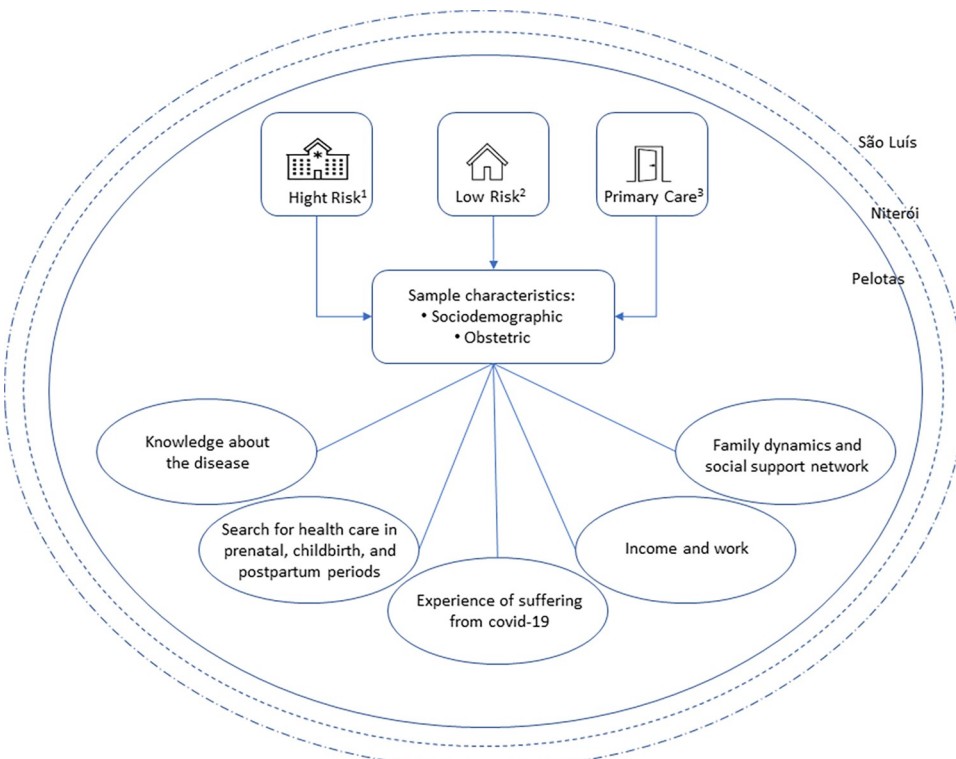

**Fig 1. Diagram of methodological steps and thematic axes obtained from data analysis.** [1]Health facilities (hospitals) providing care to high-risk pregnancies. [2]Health facilities (hospitals) providing care to low- and moderate-risk pregnancies. [3]Primary health care units for low- and moderate-risk pregnancies.

raising emotions and attitudes related to the health condition, the adequacy of new routines, changes in the forms of access to the health care network (RAS, acronym in Portuguese), and goods and services.

Despite the information mentioning pregnant women as a population at risk for infection by COVID-19 and the need for care, some of these women from São Luís believed they would not become infected and carried on their lives without further restrictions. On the contrary,

**Table 1. Knowledge about the disease, prevention and risk perception: Fragments of interviewees' statements.** São Luís-MA, Niterói-RJ and Pelotas-RS, Brazil. 2021.

| Municipality | Fragments of the interviewees' statements |
|---|---|
| **Niterói** | *"I was very afraid of getting COVID, not only for myself, but for my baby, so I avoided using public transportation, and remained indoors as much as possible." (Interview No. 01, 26-years-old, about the gestational period)* |
| **Pelotas** | *"No, I think our greatest feeling was that of uncertainty, you know, the feeling of not knowing what is going to happen, of fear, of anguish. Anyway, everything is OK, soon we will be locked up, and we do not know." (Interview No.01, 29-years-old, about the gestational period)*<br>*"Watching TV, I was scared because I was afraid of my children getting it, I was afraid of staying at the hospital and bringing the disease to my son, or my son getting it in the hospital, I was very afraid. Thank goodness nothing happened." (Interview No. 11, 18-years-old, about the gestational period)* |
| **São Luís** | *"Look! At first, I wasn't scared, right? Because I never imagined that I would catch it, (because it was) on the other side of the world, right?" (Interview No. 19, 30-years-old, about the gestational period)*<br>*"I learned about it from the newspapers, right? And… I just knew it was out there, not in Brazil. Then it came to Brazil and so on. But it was more on the television, in the newspapers. Just that." (Interview No. 04, 21-years-old, about the gestational period)* |

other women took preventive measures due to the occurrence of the disease in someone close to them.

COVID-19 infection and even measures to contain the pandemic–such as social isolation, physical distancing, and quarantine–have affected the mental health of women in several countries [25, 39]. Studies have shown an increased prevalence of minor psychological disorders during pregnancy and postpartum periods [23, 24]. High levels of stress, anxiety, and/or depression [40, 41], in addition to sadness, worries, fear, anger, irritation, frustration, loneliness, guilt, doubt, and conflict about receiving health care [34, 41] may have been a reflection of the uncertainties and difficulties experienced by these women in the context of gestating and giving birth in a pandemic caused by a new and unknown disease [5, 42]. In Brazil, the large amount of information about the disease and the dissemination of contradictory or false news, in a short period, also contributed to insecurity of these women [43].

Fragments of statements by the women interviewed are presented in Table 1 to illustrate the thematic axis.

**Search for health care in prenatal, childbirth, postpartum periods, and childcare.**
When the pandemic started, access to prenatal care was difficult in all the cities studied. At that time, COVID-19 vaccination had not yet begun. Closed routine care facilities, pregnant women referred to prenatal care in other health facilities, priority of care to those with respiratory symptoms, infection and/or transfer of professionals due to risk factors for COVID-19, and fear of commuting by public transportation led to discontinuity in prenatal care.

To mitigate the inconvenience generated by changes in the health care network, some health facilities in São Luís and Pelotas used communication technology and telemedicine for prenatal follow-up and continuity of care. Communication technologies can offer new possibilities to maintain links between healthcare professionals and patients, monitor pregnant women, especially those affected by COVID-19, avoiding discontinuity of care [44, 45]. However, the implementation strategy of these telemedicine devices must overcome the difficulties of acceptance and trust on the part of patients and ensure alternating with face-to-face visits.

A considerable number of women, mainly in Niterói and Pelotas, sought the private network to complement their visits and exams. However, in Pelotas, it was reported that even in the private network, there was discontinuity of care or replacement by remote services, by communication applications and telephone calls. Some of the women mentioned the lack of trust in the new modalities of remote assistance, and those who adopted the new modalities mentioned their previous bond in the patient-physician relationship.

A few mentioned receiving follow-ups from other professionals, such as nurses. In Niterói, some pregnant women reported visits by Community Health Agents as a different approach in pregnancy monitoring. Other professionals, such as nurses, can participate in supporting pregnant and postpartum women in terms of guidance and monitoring (in person or remotely) of pregnancy and COVID-19, increasing the bond with the team and reassuring the patients [41, 46].

The greatest difficulty reported in all municipalities studied was related to ultrasound exams, laboratory tests, and vaccination; when available, these procedures were conducted in centralized facilities, far from their area of residence. This required travelling in private vehicles which were considered safer alternatives to public transportation in terms of agglomeration and risk of infection, but resulted in unforeseen costs in their budget.

RAS must be kept open for pregnant and postpartum women, warranting prenatal care in Primary Care, facilitating access to the necessary tests, ensuring continuity and safety of care, and reinforcing humanization measures in the relationship with females, such as qualified listening, reception, bonding, and accountability.

Considering the regional inequalities in the provision of health services already pointed out by Viacava *et al.*, 2018 [47] and Albuquerque *et al.* [48, 49], in which the North and Northeast regions have less access to services than the South and Southeast regions, access barriers in four dimensions, as suggested by Oliveira *et al.*, 2019 [50], were identified in the studied municipalities as follows: i) no geographic access, since there were no health care facilities in some neighborhoods and/or localities; ii) availability of facilities, professionals and exams in an appropriate number and timely response; iii) acceptability of replacing professionals in pre-natal care, in safety procedures against COVID-19 adopted by professionals, and in remote forms of care (introduction of telemedicine devices); and iv) feasibility (payment capacity), taking into account the need to spend on private transportation and payment for visits and exams in the private sector.

Access to a maternity hospital was a relevant topic in the women's statements. This was the longest and most detailed item in the reports, in a clear demonstration of the importance attributed to this aspect. This perception stands out, especially in São Luís, given the character-istics of the women interviewed (affected by COVID-19 and with comorbidities). Barriers to admission for labor were identified since entering the service, or even before the first visit with a healthcare professional, until the moment of childbirth. A common point in statements given by pregnant women was the long waiting time during labor, indicating a delay in meet-ing their needs. At the time of delivery, the statements showed flaws in acceptability of care as it was provided. Protection measures against infection, reorganization of health services, and overburdened professionals were seen by women as obstacles to access. A feeling of "dehu-manization of the professionals" was perceived when they did not show empathy for user's complaints, or when the service did not offer the technological resources that were expected and understood as necessary for the adequate monitoring of labor. These access barriers may also be understood in terms of availability, due to the small number of maternity hospitals with exclusive beds for patients with COVID-19. Moreover, there were insufficient resources available to pay for travel to maternity hospitals, considering the long distance to be traveled and have care delivered.

In Pelotas and Niterói, access seemed to be easier. Pregnant women reported proper wel-coming and care during childbirth. They highlighted the team was realistic and relaxed some rules of social distancing to prioritize some humanized aspects, such as the presence of a com-panion during labor. This fact was very often reported by them. In Niterói, the referral of preg-nant women not affected by COVID-19 from primary care units to the studied maternity hospital attracted pregnant women from all neighborhoods of the city. Even when there was a public maternity hospital closer to their homes, in some cases, pregnant women preferred to be referred to the maternity facility under study. The fact that the State Maternity Hospital also received symptomatic pregnant women led those without symptoms and fearing infection, to seek the Municipal Maternity Hospital.

In São Luís, perhaps due to greater severity of COVID-19 in pregnant women as compared to the other two municipalities, the presence of a companion in the maternity hospitals was not allowed, and this was a source of complaint from the women. They also reported that some professionals, more sensitive to the situation of isolation of women, paid more attention to their demands and even used their own mobile phones as a means of communication between families and the parturients. The concern with aspects of humanized care for preg-nant women was highlighted by Estrela *et al.*, 2020, [51] as fundamental in mitigating women's fears and insecurities, as well as promoting well-being and preventing complications in their mental health [5, 34, 41, 42], as provided in the technical notes of the Ministry of Health regarding care for pregnant women with COVID-19 [52, 53].

The COVID-19 pandemic also impacted access to puerperal visits and routine vaccinations, leading to some discontinuity in care in all municipalities studied. Some of the women interviewed, especially in São Luís and Niterói, mentioned problems in scheduling appointments at maternity clinics and primary healthcare units, such as lack of referrals, unavailability of agenda, and late scheduling. Chisini *et al.*, 2021 [16], observed a reduction in the provision of visits throughout Brazil during 2020.

Reduced provision of antenatal visits, health infrastructure overloading, and potentially harmful policies implemented with little evidence were also reported in other countries [39]. Although the impacts of the pandemic have been greater in low- and middle-income countries [3, 54], even in developed countries, pregnant and postpartum women have experienced interruptions in access to obstetric health care. The perceived quality of obstetric care was negatively influenced by interruption of services, cancellation of face-to-face appointments, replacement by virtual and/or telephone consultations, and exclusion of the companion during hospitalization for childbirth [27]. In addition to problems in delivering health services during pregnancy, childbirth, and puerperium, there were challenges related to transportation, social isolation measures, or fear of being infected when attending health services [55].

Some fragments of statements by women interviewed are shown in Table 2 to illustrate the thematic axis.

**Table 2. Search for health care in prenatal, childbirth and postpartum periods: Fragments of the interviewees' statements.** São Luís-MA, Niterói-RJ and Pelotas-RS, Brazil. 2021.

| Municipality | Fragments of the interviewees' statements |
|---|---|
| Niterói | *"The prenatal care, which are the family centers, I have nothing to complain about them, they are great, attentive, cautious, they always make appointments month after month, I did all the free consultations quarterly, with results on time, in all the appointments I was assisted by the wonderful doctor who has there, thank goodness." (Interview No. 02, 22-years-old, about the gestational period, UBS)* <br> *"Oh, you have to do the ultrasound", they gave me the paper and told me to go there. "Go to the clinic, go somewhere else, and so on". (Interview No. 03, 17-years-old, about the gestational period, UBS)* <br> *"I had to pay for the 'ultras' as it takes a long time to get the referral, so I had to pay to be able to advance the process for the doctor." (Interview No. 04, 30-years-old, about the gestational period, UBS)* <br> *"He was born, then they took him to take the vaccine, cleaned him, and immediately brought him to me… Then my mother took him for about 15 minutes for me to finish having the stitches, and taking a shower, the nurses helped me take a shower quickly, then immediately he was breastfed, and he had the first contact with me… Then I was with him until discharge." (Interview No. 05, 23-years-old, about the postpartum period, MRH)* |
| Pelotas | *"Yes, appointments, some tests, ultrasounds, I had to pay for them all." (Interview No. 08, 34-years-old, about the gestational period)* |
| São Luís | *"It was then that I started doing prenatal care at the health care facility because the pandemic was already there, then before I did my prenatal care as it was risky because of diabetes, they were not attending people there because of that, right? Then the nurse came and said: 'Now you are going to come here at the facility to take these vaccines and talk to us online on the cell phone,' so that was it." (Interview No. 09, 33-years-old, about pregnancy, UBS)* <br> *"Yes, for sure (the pandemic affected my prenatal care). Because we have to have the follow-up, and everything must be quite right. And then we couldn't go to the health facility, because there were, especially here, the facility in (name of the neighborhood), there were a lot of people. So, we also didn't have that ease of moving around, going to other places, also because of… this thing that we don't know who has it, who doesn't, you know? And for, for those who are pregnant, ah, it's much easier to catch, because we have (low) immunity, right?" (Interview No. 10, 42-years-old, about pregnancy, UBS)* <br> *"Oh no, I didn't have (a companion). My mother, who went with me every time, they did not let her in. She went (and stayed) outside the maternity hospital." (Interview No. 16, 17-years-old, about childbirth, MAR)* |

MRH: low- and moderate risk maternity hospital. MAR: high-risk maternity hospital. UBS: primary care unit. (All acronyms are in Portuguese).

**Experience of suffering from COVID-19.**   In the cities studied, the experience of illness among women was different. In São Luís, 14 were hospitalized, five of them at the ICU. In Niterói, the only pregnant woman who reported having had COVID-19 during pregnancy had mild symptoms. In Pelotas, most women had mild symptoms, such as tiredness, loss of smell, and runny nose. Only two women experienced shortness of breath, but this did not result in hospitalization. Although pregnant and postpartum women are not at greater risk of COVID-19 infection than other women, symptomatic and poorer women may have more adverse outcomes compared to non-pregnant women [39].

All women interviewed expressed the perception of inability to avoid infection as one of their frustrations, since they complied with safety recommendations, such as social isolation from family members, restrictive measures for home visits, use of masks, alcohol-based hand rub, bathing and changing clothes when arriving from external environments and being careful to avoid sharing utensils. Some women mentioned they had expectations about the moment of delivery, which were altered by the circumstances. Shuman *et al.*, 2022 [56], commented on the frustration of North American women in the idealized experiences of pregnancy and childbirth that were very different from those experienced during the pandemic. In contexts of greater social vulnerability, pregnant and postpartum women felt helpless, and the joyful event of pregnancy turned into fear and stress [41].

Many Brazilian women were infected during pregnancy, childbirth, and postpartum periods, with a high impact on maternal morbidity and mortality. In this sense, the pandemic may impact the achievement of Sustainable Development Goals. Before the pandemic, Brazil was expected to achieve a 46.6% reduction in the maternal mortality ratio by 2030 [57]. However, Brazil had the highest rates of maternal mortality due to COVID-19, worldwide [21, 22]. Limited resources for provision of health care [54], lack of women-centered care [19], inefficiency of health systems, and inability to adequately manage the pandemic [58] contributed to worsening of the adverse impacts of the pandemic in reproductive health; and emphasized the need for appropriate measures of adequate antenatal and postnatal care.

Some fragments of statements by the interviewed women are displayed in Table 3 to illustrate the thematic axis.

**Income and work.**   The work status of pregnant women was greatly affected by the pandemic in all municipalities. The preventive removal of those who had contact with the public in trade and service activities, the low customer demand, and the removal due to illness of

**Table 3. Experience of suffering from COVID-19: Fragments of the interviewees' statements.** São Luís-MA, Niterói-RJ and Pelotas-RS, Brazil. 2021.

| Municipality | Fragments of the interviewees' statements |
|---|---|
| Niterói | *"I didn't look for anything and I didn't even go to the health center, because, as my husband had it and he had to be isolated and stay here, as we are poor, there's no way to isolate ourselves, the tendency in our home was for everyone to catch it, but God was so good that only me, my oldest son and his father had it. The little ones didn't catch it."* (Interview No. 06, 31-years-old, about the postpartum period) |
| São Luís | *"Look, I felt a lot of weakness. I was no longer able to move. I was quite. . . just throwing up. I felt short of breath. I felt all the symptoms, all of them!"* (Interview No. 12, 24-years-old, about the period of ICU stay during pregnancy, MAR)<br>*"When did I return? From COVID. . . I waited. . . I decided to do one more isolation at home. So, as there was a spare room in my mother-in-law's house and I wasn't sure if she had it [. . .]. So, I said like this: 'No. I'm going to stay in the isolated room,' And I stayed for about ten days in the isolation that I decided to do myself."* (Interview No. 01, 28-years-old, about the gestational period)<br>*"Yes, I could breastfeed her, because it didn't imply anything. . . COVID with breast milk."* (Interview No. 13, 28-years-old, about the postpartum period) |

MAR: high-risk maternity. UBS: Primary Care Unit (all acronyms in Portuguese).

them and/or their spouses, significantly modified the dynamics of work and income in the families. Most women sought emergency aid (Federal) as a way to supplement the income lost because of the pandemic. In Niterói, the municipal government developed a broad income guarantee policy [59] including emergency aid with municipal resources.

There were complaints from women suffering from COVID-19, who resorted to Social Security disease aid, about the delay in granting or not receiving the benefit. The income situation is sensitive as a social health determinant, especially during COVID-19 pandemic [60]. Other countries also reported women had greater loss of income due to the pandemic than men; and even greater work overload due to the accumulation of their work activities and child care [39]. International agencies recommended the development of employment and income maintenance policies during the pandemic, as observed in some countries [61]. When resuming comprehensive care, it is necessary to consider the income of women, especially the most vulnerable, pregnant women, and those with young children.

Table 4 shows some fragments of statements by the interviewed women to illustrate the thematic axis.

**Family dynamics and social support network.** As to the social support network, most women reported having the presence of a partner or family member during pregnancy, childbirth, and postpartum periods. Despite reporting this social contact, they mentioned a routine making significant use of communication technologies (phone calls, video calls, and messages through communication applications), able to create different support from what was expected for this stage of life, but recognized as essential during pregnancy and the birth of babies. The distance from family and friends, despite being seen as necessary, was a gap identified by the pregnant women that left them helpless from the daily help they expected.

The maintenance of social support networks plays an important role in daily life and household and work tasks, as well as in well-being of women, linking family and friends according to availability and safety conditions. Social support is defined as a buffer against stressors, and it has emotional, informational, and instrumental aspects, and is also a protective factor against postpartum depression. Communication technologies and social media can play an important role when social isolation is established [62].

To illustrate the thematic axis, Table 5 shows some fragments of statements made by the interviewed women.

## Limitations of the study

Comparisons among the three municipalities should be analyzed with caution, given that in São Luís only women diagnosed with COVID-19, with a history of admission to a reference hospital for high-risk pregnancies were included, while in the other municipalities, women

**Table 4. Income and work: Fragments of the interviewees' statements.** São Luís-MA, Niterói-RJ and Pelotas-RS, Brazil. 2021.

| Municipality | Fragments of the interviewees' statements |
|---|---|
| Niterói | *"I used to work with nails, right, there in downtown Niterói and because of the pandemic, isolation, people could not see each other, and it affected me financially." (Interview No. 02, 22-years-old, about the gestational period)* <br> *"I was at home because as soon as I got pregnant, the pandemic started. I had to be in isolation. I worked there for 2 weeks and then it closed. I work in daycare and then it had to close." (Interview No. 06, 31-years-old, about the gestational period)* |
| São Luís | *"But my mother has a store and she needed to go back to open the store, right? Because that's where she makes her living, we pay the bills by her sales, and... I had to go to my boyfriend's house. And that's when I ended up getting it. Because he also needed to buy construction stuff and he had to go out." (Interview No. 01, 28-years-old, gestational period).* |

**Table 5. Family dynamics and social support network: Fragments of the interviewees' statements.** São Luís-MA, Niterói-RJ and Pelotas-RS, Brazil. 2021.

| Municipality | Fragments of the interviewees' statements |
|---|---|
| Niterói | *"Regarding the pandemic, it was very difficult, very difficult, due to the isolation, the distance from my family and my mother. I only came here with my husband, I'm young, right, my first pregnancy went ahead, so it was very difficult, I felt distressed, very sad, and especially impotent, I couldn't do anything to solve it." (Interview No. 02, 22-years-old, about the gestational period)* |
| Pelotas | *"Yeah, I had more support from my husband, my daughter, who were in the house with me, that whole thing, and from friends over the phone." (Interview No. 09, 34-years-old about the gestational period)*<br>*"Look, I've always been a party girl. I've always liked people a lot, agglomerations, everything that can't be done now. In this case, but in the current situation, I thought about doing it closer to the birth and doing a "tea drive" in which people just pass by and take a souvenir and with a mask, with all the precautions, but not in person, right, even because there is no way." (Interview No. 03, 27-years-old, about the gestational period)* |
| São Luís | *"So, it changed on my husband's part, who changed a lot. We kept our distance because of the disease" (Interview No. 05, 40-years-old, about the gestational period)*<br>*"At the time, [he] couldn't be there. Yeah. . .my family either. To this day, there are people in my family who still don't know her [daughter]. Because. . . to prevent many people from coming here. Although we go out, there are people who haven't had it yet, and we don't know if they still have the symptoms. So a lot of people still don't know her, and she's already grown up. On the part of her father's family, many people still don't know her. Her grandmother, from her father's side, still doesn't know her." (Interview No. 03, 32-years-old, about the postpartum period, MAR)*<br>*"No. The other [son, the eldest] only came after. . . he came when he was two months old, he came to meet his little brother. He stayed with grandma. From his father's side." (Interview No. 06, 29-years-old, about the postpartum period)*<br>*". . . only my husband that I could count on. Only him. The people who were helping me, like my family, everything far, far away. Then who I have to count on are my children here. The people from the church and God, right?" (Interview No. 11, 40-years-old, about the postpartum period)* |

MAR: high-risk maternity (acronym in Portuguese).

who attended medium-complexity hospitals were also included. On the other hand, this strategy allowed us to explore the differences among women belonging to different regions of the country and with different demographic characteristics. In general, the women selected, especially in São Luís, were of low- and middle- income, which limits the external validity of our findings, but we still looked for a considerable range of socioeconomic and demographic characteristics. Another limitation was that we conducted some interviews via virtual platforms. At first, we thought that the interview conducted through virtual platforms could make it difficult to approach sensitive topics, such as the fear of contracting the disease and/or the difficulties encountered in accessing health services. Another difficulty could be the interruptions caused by the low quality of the users' internet connection. However, this way of conducting interviews reduced risks for women who expressed they felt safe and respected by the researchers, and did not impair the quality of the interview. Besides, for ethical reasons, we could not make public the database containing the transcription of all the speeches of the women interviewed, which is a limitation for open science. However, we transcribed several excerpts of statements into the manuscript, selecting representative statements from the sample, and pointing out agreements and disagreements, when present. In addition, some de-identified data may be made available upon request.

## Final considerations

The experience of the pandemic by women was different in the three municipalities. The pandemic had a negative impact on access to prenatal care, humanized childbirth, puerperal visits and the routine vaccination service, which are aspects of primary care that require improved planning in the three levels of health management.

Our participants stated they felt safer and had access to the services more often when provided with information. The media played a very important role in conveying information and in combatting fake news or information without scientific evidence. It is necessary to invest in reliable sources of information, based on the best scientific evidence, delivered on time, to better guide professionals and the population.

The pressures naturally associated with pregnancy have been greatly amplified by the changes brought about by the pandemic. Careful listening by professionals and mental health support can prevent more serious conditions, such as depression.

The pandemic contributed to worsening of the population's social and economic vulnerability. Policies to maintain employment and income are essential to mitigate such vulnerability and improve the population's adoption of prevention and social isolation measures.

Although most of the women interviewed had some type of family support during the period, social isolation was responsible for disrupting the social support network. The maintenance of these networks plays an important role in the daily life and household and work tasks, as well as in well-being of women, linking family and friends according to availability and safety conditions. Communication technologies played an important role in the reorganization of the social support network.

## Acknowledgments

The authors would like to thank the women who participated in this research.

## Author Contributions

**Conceptualization:** Zeni Carvalho Lamy, Erika Barbara Abreu Fonseca Thomaz, Aluísio Gomes da Silva-Junior, Gisele Caldas Alexandre, Maria Teresa Seabra Soares de Britto e Alves, Ruth Helena de Souza Britto Ferreira de Carvalho, Sandro Schreiber de Oliveira, Tatiana Raquel Selbmann Coimbra, Lely Stella Guzman-Barrera.

**Data curation:** Zeni Carvalho Lamy, Aluísio Gomes da Silva-Junior, Ruth Helena de Souza Britto Ferreira de Carvalho, Sandro Schreiber de Oliveira.

**Formal analysis:** Zeni Carvalho Lamy, Erika Barbara Abreu Fonseca Thomaz, Aluísio Gomes da Silva-Junior, Gisele Caldas Alexandre, Maria Teresa Seabra Soares de Britto e Alves, Ruth Helena de Souza Britto Ferreira de Carvalho, Letícia Oliveira de Menezes, Sandro Schreiber de Oliveira, Maurício Moraes, Yasmim Bezerra Magalhães.

**Funding acquisition:** Zeni Carvalho Lamy, Aluísio Gomes da Silva-Junior, Sandro Schreiber de Oliveira, Tatiana Raquel Selbmann Coimbra, Lely Stella Guzman-Barrera.

**Investigation:** Zeni Carvalho Lamy, Erika Barbara Abreu Fonseca Thomaz, Aluísio Gomes da Silva-Junior, Gisele Caldas Alexandre, Maria Teresa Seabra Soares de Britto e Alves, Ruth Helena de Souza Britto Ferreira de Carvalho, Letícia Oliveira de Menezes, Sandro Schreiber de Oliveira, Maurício Moraes, Yasmim Bezerra Magalhães.

**Methodology:** Zeni Carvalho Lamy, Erika Barbara Abreu Fonseca Thomaz, Aluísio Gomes da Silva-Junior, Gisele Caldas Alexandre, Maria Teresa Seabra Soares de Britto e Alves, Ruth Helena de Souza Britto Ferreira de Carvalho, Letícia Oliveira de Menezes, Sandro Schreiber de Oliveira, Maurício Moraes, Yasmim Bezerra Magalhães, Tatiana Raquel Selbmann Coimbra, Lely Stella Guzman-Barrera.

**Project administration:** Zeni Carvalho Lamy, Aluísio Gomes da Silva-Junior, Sandro Schreiber de Oliveira, Tatiana Raquel Selbmann Coimbra, Lely Stella Guzman-Barrera.

**Resources:** Tatiana Raquel Selbmann Coimbra, Lely Stella Guzman-Barrera.

**Software:** Tatiana Raquel Selbmann Coimbra, Lely Stella Guzman-Barrera.

**Supervision:** Zeni Carvalho Lamy, Aluísio Gomes da Silva-Junior, Gisele Caldas Alexandre, Ruth Helena de Souza Britto Ferreira de Carvalho, Sandro Schreiber de Oliveira, Maurício Moraes, Tatiana Raquel Selbmann Coimbra, Lely Stella Guzman-Barrera.

**Validation:** Zeni Carvalho Lamy, Erika Barbara Abreu Fonseca Thomaz, Aluísio Gomes da Silva-Junior, Gisele Caldas Alexandre, Maria Teresa Seabra Soares de Britto e Alves, Ruth Helena de Souza Britto Ferreira de Carvalho, Letícia Oliveira de Menezes, Sandro Schreiber de Oliveira, Maurício Moraes, Yasmim Bezerra Magalhães, Tatiana Raquel Selbmann Coimbra, Lely Stella Guzman-Barrera.

**Visualization:** Zeni Carvalho Lamy, Erika Barbara Abreu Fonseca Thomaz, Aluísio Gomes da Silva-Junior, Gisele Caldas Alexandre, Maria Teresa Seabra Soares de Britto e Alves, Ruth Helena de Souza Britto Ferreira de Carvalho, Letícia Oliveira de Menezes, Sandro Schreiber de Oliveira, Maurício Moraes, Yasmim Bezerra Magalhães, Tatiana Raquel Selbmann Coimbra.

**Writing – original draft:** Zeni Carvalho Lamy, Erika Barbara Abreu Fonseca Thomaz, Aluísio Gomes da Silva-Junior, Gisele Caldas Alexandre, Maria Teresa Seabra Soares de Britto e Alves, Ruth Helena de Souza Britto Ferreira de Carvalho, Letícia Oliveira de Menezes, Sandro Schreiber de Oliveira, Maurício Moraes, Yasmim Bezerra Magalhães, Tatiana Raquel Selbmann Coimbra, Lely Stella Guzman-Barrera.

**Writing – review & editing:** Zeni Carvalho Lamy, Erika Barbara Abreu Fonseca Thomaz, Aluísio Gomes da Silva-Junior, Gisele Caldas Alexandre, Maria Teresa Seabra Soares de Britto e Alves, Ruth Helena de Souza Britto Ferreira de Carvalho, Letícia Oliveira de Menezes, Sandro Schreiber de Oliveira, Maurício Moraes, Yasmim Bezerra Magalhães, Tatiana Raquel Selbmann Coimbra, Lely Stella Guzman-Barrera.

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
