## [Decision Letter · Decision Letter 0]

12 Dec 2022

PONE-D-22-19394Experiences of women in prenatal, childbirth and postpartum health care during the COVID-19 pandemic in selected cities in Brazil: the resignification of the experience of pregnancy and giving birthPLOS ONE

Dear Dr. Thomaz,

Thank you for submitting your manuscript to PLOS ONE. After careful consideration, we feel that it has merit but does not fully meet PLOS ONE’s publication criteria as it currently stands. Therefore, we invite you to submit a revised version of the manuscript that addresses the points raised during the review process.

We look forward to receiving your revised manuscript.

Kind regards,

Ivan Filipe de Almeida Lopes Fernandes, Ph.D.

Academic Editor

PLOS ONE

https://journals.plos.org/plosone/s/fileid=ba62/PLOSOne_formatting_sample_title_authors_affiliations.pdf.

3. You indicated that you had ethical approval for your study. In your Methods section, please ensure you have also stated whether you obtained consent from parents or guardians of the minors included in the study or whether the research ethics committee or IRB specifically waived the need for their consent.

“The authors are grateful for the technical and financial support of the Bill and Melinda Gates Foundation [INV-017424], World Health Organization (WHO) and Pan American Health Organization (PAHO) [CON20-00012173] - ZCL, EBAFT, AGSJ, GCA, MTSSBA, RHSBFC, LOM, SSO, MM, YBM, TRSC, BBR, LSGB. Also, the National Council for Scientific and Technological Development (Conselho Nacional de Desenvolvimento Científico e Tecnológico – CNPq acronym in Portuguese) [processes 306592/2018-5 (EBAFT), 314939/2020-2 (ZCL), 311479/2020-2 (MRSSBA) and 308917/2021-9 (EBAFT)] and the Coordination for the Improvement of Higher Education Personnel (Coordenação de Aperfeiçoamento de Pessoal de Nível Superior – CAPES acronym in Portuguese) [finance code 001]  (EBAFT, MTSSBA, RHSBFC, ZCL) for support for scientific publication.”

Additional Editor Comments:

The theme of the article is relevant.

We would recommend a more accurate review of the language as well as appropriate sources referencing, as suggested by the reviewer.

In addition, we strongly recommend that you create a limitations section that shows the scope of the conclusions made in the study.

Finally, we recommend a full review of Table 1 with a better and more appropriate presentation. It is of fundamental importance that the table (or tables) in the manuscript carry information that is easily recognized by the reader, making the transmission of data more fluid and efficient.

Reviewers' comments:

Reviewer's Responses to Questions

**Comments to the Author**

1. Is the manuscript technically sound, and do the data support the conclusions?

Reviewer #1: Yes

Reviewer #2: Partly

2. Has the statistical analysis been performed appropriately and rigorously? 

Reviewer #1: N/A

Reviewer #2: N/A

3. Have the authors made all data underlying the findings in their manuscript fully available?

Reviewer #1: No

Reviewer #2: No

4. Is the manuscript presented in an intelligible fashion and written in standard English?

Reviewer #1: Yes

Reviewer #2: No

5. Review Comments to the Author

Reviewer #1: The article was written in standard English and presented in a well-structured and intelligent fashion. It was qualitative, exploratory research that used individual semi-structured interviews for data collection. Content analysis of the results were presented through thematic modality and thus no need for statistical analysis.

The authors need to explain if the ethical approval approved by the Research Ethics Committee of the University Hospital

221 of the Federal University of Maranhão covers all the three (3) municipals study areas.

The authors did not provide the excerpts of all the forty-six (46) study participants. Therefore, they need to explain the reasons for the restrictions of the remaining excerpts or should provide contact information for data access if there is privacy of the subjects cannot be completely protected.

Reviewer #2: The manuscript describes an important and relevant research topic. It is a qualitative paper, so no statistical analysis is needed. However, it is important to review the language.

Lines 112-119 need to be re-written due to similarity with https://linkinghub.elsevier.com/retrieve/pii/S2667193X22000564 and https://pubmed.ncbi.nlm.nih.gov/34713913/. Please make sure both papers are correctly paraphrased, cited, or content removed.

The same happens on lines 398-400; 406-410; 442-443; It needs to be re-written since the text corresponds to the main source wording without rephrasing.

The software used to assess similarity is Turnitin.

Can Table 1 be split into different tables or use a diagram to present some of this information?

Please include a "Limitations" section in the paper.

"The stress naturally generated in pregnancy (...)" - since stress is a concept and this was not evaluated in this study, I suggest replacing this word in the final considerations in order not to be misleading.

"The availability of information about the disease and its prevention provided more tranquility and greater adherence" - Please re-write to avoid suggesting causal effects. "Our participants referred to feel safer and also to access the services more often when provided with information," for example, would accurately express that this is a social phenomenon inferred from their interviews.

6. PLOS authors have the option to publish the peer review history of their article (what does this mean?). If published, this will include your full peer review and any attached files.

Reviewer #1: **Yes: **Ibrahim Abbas Lawal

Reviewer #2: **Yes: **Heloísa Garcia Claro

---

## [Author Response · Author response to Decision Letter 0]

16 Feb 2023

PLOS ONE

PONE-D-22-19394

Experiences of women in prenatal, childbirth, and postpartum care during the COVID-19 pandemic in selected cities in Brazil: the resignification of the experience of pregnancy and giving birth

Dear Academic Editor and Reviewers,

Thank you for the careful evaluation of the manuscript referred to above. After careful reading, we considered all suggestions made and responded item by item in this document. We are resubmitting the manuscript to Plos One, addressing the points raised during the review process. In addition to the 'Rebuttal letter', we have also included a 'Revised Manuscript with Track Changes' and an unmarked version of the paper without tracked changes, under the instructions received.

Thank you very much for the opportunity to review the work.

Kind regards,

The Authors

1. Please ensure that your manuscript meets PLOS ONE's style requirements, including those for file naming. The PLOS ONE style templates can be found at https://journals.plos.org/plosone/s/file?id=wjVg/PLOSOne_formatting_sample_main_body.pdf and https://journals.plos.org/plosone/s/fileid=ba62/PLOSOne_formatting_sample_title_authors_affiliations.pdf. 

Authors' response:

Thank you very much. We reviewed according to PLOS ONE's style requirements, including those for file naming and templates.

2. Please provide additional details regarding participant consent. In the ethics statement in the Methods and online submission information, please ensure that you have specified (1) whether consent was informed and (2) what type you obtained (for instance, written or verbal, and if verbal, how it was documented and witnessed). If your study included minors, state whether you obtained consent from parents or guardians. If the need for consent was waived by the ethics committee, please include this information. If you are reporting a retrospective study of medical records or archived samples, please ensure that you have discussed whether all data were fully anonymized before you accessed them and/or whether the IRB or ethics committee waived the requirement for informed consent. If patients provided informed written consent to have data from their medical records used in research, please include this information.

Authors' response:

Thank you. A written and signed informed consent was obtained from each participant, and the study protocol was approved by the Universidade Federal do Maranhão, Universidade Católica de Pelotas, and the Pan American Health Organization. The following text has been added:

“The research was approved by the Research Ethics Committee of the University Hospital of the Universidade Federal do Maranhão (CAAE 35645120.9.0000.5086, approved on August 25th, 2020, and amended on November 27th, 2020, to include Pelotas and Niterói). Besides, the research was further approved by the Research Ethics Committees of the Universidade Católica de Pelotas (CAAE 38281820.3.0000.5339, approved on November 9th, 2020) and the Pan American Health Organization (Ref. number PAHOERC.0260.02 on October 19th, 2020), in compliance with the Resolutions 466/12 and 510/16 of the National Health Council. All study participants signed an informed consent form in two copies - one was delivered to each participant, and the other kept by the project team. A written assent was signed by parents or guardians in case of adolescent participants. Funders did not interfere in the methodology or any other step that could influence the results or conclusions of the study. The interviewees were granted anonymity. The names were replaced by codes that started with the first letter of the municipality (N – Niterói; P – Pelotas; and S – São Luís), followed by the serial number of the interview, age of the woman interviewed, and period the recorded text referred to (pregnancy, childbirth or postpartum). When the interviewee referred to the care received in the health care network, the setting of care (MRH - low- and moderate-risk maternity; MAR - high-risk maternity; and UBS – primary care unit, acronyms in Portuguese) was inserted at the end of the coding.”

Proofs of the ethics committee approvals are attached. 

3. You indicated that you had ethical approval for your study. In your Methods section, please ensure you have also stated whether you obtained consent from parents or guardians of the minors included in the study or whether the research ethics committee or IRB specifically waived the need for their consent.

Authors' response:

Many thanks for the careful review of ethical issues involving the study population. Researchers obtained written and signed consent from parents or guardians in cases of adolescent pregnancies - after clarifying the research questions, objectives, methods, risks, benefits, and other aspects of the study. This information was added to the text, as mentioned in the previous item.

“The authors are grateful for the technical and financial support of the Bill and Melinda Gates Foundation [INV-017424], World Health Organization (WHO), and Pan American Health Organization (PAHO) [CON20-00012173] - ZCL, EBAFT, AGSJ, GCA, MTSSBA, RHSBFC, LOM, SSO, MM, YBM, TRSC, LSGB. Also, the National Council for Scientific and Technological Development (Conselho Nacional de Desenvolvimento Científico e Tecnológico – CNPq acronym in Portuguese) [processes 306592/2018-5 (EBAFT), 314939/2020-2 (ZCL), 311479/2020-2 (MRSSBA) and 308917/2021-9 (EBAFT)] and the Coordination for the Improvement of Higher Education Personnel (Coordenação de Aperfeiçoamento de Pessoal de Nível Superior – CAPES acronym in Portuguese) [finance code 001] (EBAFT, MTSSBA, RHSBFC, ZCL) for support for scientific publication.”

Please state what role the funders took in the study. If the funders had no role, please state: "The funders had no role in study design, data collection, and analysis, decision to publish, or preparation of the manuscript."

Author's response:

Dear Editor and Reviewers, the text referring to the funders must be edited, according to the text below:

“The authors are grateful for the technical and financial support of the Bill and Melinda Gates Foundation [INV-017424], World Health Organization (WHO), Pan American Health Organization (PAHO) [ZCL, EBAFT, AGSJ, GCA, MTSSBA, RHSBFC, LOM, SSO, MM, YBM, TRSC, LSGB], National Council for Scientific and Technological Development (CNPq, acronym in Portuguese) [processes 306592/2018-5 (EBAFT), 314939/2020-2 (ZCL), 311479/2020-2 (MRSSBA), and 308917/2021-9 (EBAFT)], and the Coordination for the Improvement of Higher Education Personnel (CAPES, acronym in Portuguese) [finance code 001] (EBAFT, MTSSBA, RHSBFC, ZCL) for supporting scientific publication. The funders had no role in study design, data collection, and analysis, decision to publish, or preparation of the manuscript.”

At the end of the ethical considerations, we also informed the funders did not interfere in the methodology or any other step that could influence the results or conclusions of the study. 

If there are ethical or legal restrictions on sharing a de-identified data set, please explain the restrictions in detail (e.g., data contain potentially identifying or sensitive patient information) and who has imposed them (e.g., a Research Ethics Committee or Institutional Review Board, etc.). Please also provide non-author contact information (phone/email/hyperlink) for a data access committee, ethics committee, or other institutional body to which data requests may be sent. If applicable, please also provide any necessary information which interested researchers would need when requesting access to data in order to obtain the minimal data set for your study.

Authors' response:

Dear Editor, we did not make available all the data (transcription of all speeches), but we have inserted several excerpts from the speeches of the women (de-identified) throughout the manuscript. As it would not be possible to insert the speeches of all the women in the manuscript, we selected representative excerpts from the identified thematic axes; hence, the statements that support our findings and conclusions were provided in the manuscript.

Because this is a qualitative research, with deep interviews, addressing sensitive issues from the point of view of identifying women, providing the entire transcription of the interviews can violate the ethical precepts of guaranteeing the secrecy and privacy of participants. In this way, we will only be able to make the data available upon request through e-mail to a non-author contact: Sandra Santos, Pan-American Health Organization – PAHO, E-mail: sandra@paho.org, Phone: +55 61 32519513).

Additional Editor Comments:

The theme of the article is relevant. We would recommend a more accurate review of the language as well as appropriate sources referencing, as suggested by the reviewer.

Authors' response:

Thank you very much for the comment. A professional language specialist who translates and revises for the Pan American Health Organization reviewed the manuscript. The service declaration is attached.

In addition, we strongly recommend that you create a limitations section that shows the scope of the conclusions made in the study.

Authors' response:

We have added a limitations section to the Discussion, as shown below. Thank you for the suggestion. 

“Comparisons among the three municipalities should be analyzed with caution, given that in São Luís only women diagnosed with COVID-19, with a history of admission to a reference hospital for high-risk pregnancies were included, while in the other municipalities, women who attended medium complexity hospitals were also included. On the other hand, this strategy allowed us to explore the differences among women belonging to different regions of the country and with different demographic characteristics. In general, the women selected, especially in São Luís, were of low- and middle-income, which limits the external validity of our findings, but we still looked for a considerable range of socioeconomic and demographic characteristics. Another limitation was that we conducted some interviews via virtual platforms. At first, we thought that the interview conducted through virtual platforms could make it difficult to approach sensitive topics, such as the fear of contracting the disease and/or the difficulties encountered in accessing health services. Another difficulty could be the interruptions caused by the low quality of the users' internet connection. However, this way of conducting interviews reduced risks for women who expressed they felt safe and respected by the researchers, and did not impair the quality of the interview.”

Finally, we recommend a full review of Table 1 with a better and more appropriate presentation. It is of fundamental importance that the table (or tables) in the manuscript carry information that is easily recognized by the reader, making the transmission of data more fluid and efficient.

Authors' response:

We followed the reviewers' suggestions. We split Table 1 into 5 different tables. We also edited tables in a better and more appropriate presentation. In addition, we created a Figure with a diagram referring to the five thematic axes of the study. In this way, the visualization of data is more fluid, efficient, and easier to understand.

Reviewers' comments:

Reviewer's Responses to Questions

Comments to the Author

1. Is the manuscript technically sound, and does the data support the conclusions?

Reviewer #1: Yes

Reviewer #2: Partly

Authors' response:

We appreciated the assessment. We have written a section on the limitations of the study to help analyze the results and conclusions in light of the challenges of the study.

2. Has the statistical analysis been performed appropriately and rigorously?

Reviewer #1: N/A

Reviewer #2: N/A

Authors' response:

Since this is a qualitative approach study, there is no room for statistical analysis of the data. Content analysis was performed, referred to in the text. 

3. Have the authors made all data underlying the findings in their manuscript fully available?

Reviewer #1: No

Reviewer #2: No

Authors' response:

Dear Editor, since this is a qualitative research, the calculation of means, medians, and other statistical measures does not apply to these data. On the other hand, as mentioned on the previous page, we do not have ethical authorization to widely make available the transcription of all qualitative interviews carried out (full open access). In this way, we will only be able to make the full transcriptions available upon request by e-mail to a non-author contact: Sandra Santos, Pan-American Health Organization – PAHO, E-mail: sandra@paho.org, Phone: +55 61 32519513).

4. Is the manuscript presented in an intelligible fashion and written in standard English?

Reviewer #1: Yes

Reviewer #2: No

Authors' response:

Thank you very much for the comment. A professional language specialist who translates and revises for the Pan American Health Organization reviewed the manuscript. The service declaration is attached.

5. Review Comments to the Author (Please use the space provided to explain your answers to the questions above. You may also include additional comments for the author, including concerns about dual publication, research ethics, or publication ethics. (Please upload your review as an attachment if it exceeds 20,000 characters))

Reviewer #1: The article was written in standard English and presented in a well-structured and intelligent fashion. It was qualitative, exploratory research that used individual semi-structured interviews for data collection. Content analysis of the results were presented through thematic modality and thus no need for statistical analysis.

The authors need to explain if the ethical approval approved by the Research Ethics Committee of the University Hospital 221 of the Federal University of Maranhão covers all the three (3) municipals study areas.

Authors' response:

Thank you very much for the comment. The study was approved by the UFMA Ethics Committee, on August 25th, 2020, to be carried out in São Luís. We decided to expand the study area, and requested a new ethical evaluation from the UFMA Committee (an amendment to the original ethical approval) including permission to expand the study to Pelotas (RS) and Niterói (RJ). The new ethical approval was issued on November 27th, 2022. In addition, we obtained further approvals from the Universidade Católica de Pelotas on November 9th, 2020, and from the Research Ethics Committee of the Pan American Health Organization on October 19th, 2020. Thus, we have the authorization to carry out the study in the three cities – São Luís, Pelotas, and Niteroi. We have included this piece of information in the manuscript and attached ethics approval documents at the end of this Rebuttal Letter.

The authors did not provide the excerpts of all the forty-six (46) study participants. Therefore, they need to explain the reasons for the restrictions of the remaining excerpts or should provide contact information for data access if there is privacy of the subjects cannot be completely protected.

Authors' response:

Thank you very much for your comments. In fact, we did not include excerpts of speeches of all 46 women participating in the study. We analyzed all responses and selected fragments that represented each thematic axis. The selection aimed to address statements by different women, with diverse characteristics, which were also representative of the entire sample. By and large, the answers were repeated or complemented each other. When divergent, were presented both speeches with contradictory views, whereas in cases of agreement, the excerpts accounting more for the entire sample were selected. Due to ethical and legal issues, the full-text transcriptions can be provided upon formal request to a non-author contact: Sandra Santos – Pan-American Health Organization – PAHO; E-mail: sandra@paho.org; Phone: +55 61 32519513.

Reviewer #2: The manuscript describes an important and relevant research topic. It is a qualitative paper, so no statistical analysis is needed. However, it is important to review the language.

Author's response:

Thank you. The manuscript is resubmitted after language review. 

Lines 112-119 need to be re-written due to similarity with https://linkinghub.elsevier.com/retrieve/pii/S2667193X22000564 and https://pubmed.ncbi.nlm.nih.gov/34713913/. Please make sure both papers are correctly paraphrased, cited, or content removed.

Authors' response:

Thank you very much for this suggestion. We wrote the sentence again to avoid similarity/ equality. The new sentence is transcribed below:

“Three situations were identified as key aspects responsible for maternal deaths in Brazil. First, delay in identifying and testing pregnant women with COVID-19 symptoms. Second, delay in hospital admission of women diagnosed with COVID-19. Third, delay in providing timely treatment at intensive care units (ICU) [19]. Therefore, appropriate management of pregnancy during the COVID-19 pandemic was a tough issue [20].”

The same happens on lines 398-400; 406-410; 442-443; It needs to be re-written since the text corresponds to the main source wording without rephrasing.

The software used to assess similarity is Turnitin.

Authors' response:

We appreciate the careful verification of similarities in our manuscript concerning the literature. We rewrote the text fragments as described below:

Lines 398-400: In addition to problems in delivering health services during pregnancy, childbirth, and puerperium, there were challenges related to transportation, social isolation measures, or fear of being infected when attending health services [55].

Lines 406-410: Although pregnant and postpartum women are not at greater risk of COVID-19 infection than other women, symptomatic and poorer women may have more adverse outcomes compared to non-pregnant women [39].

Lines 442-443: Other countries also reported women had greater loss of income due to the pandemic than men; and even greater work overload due to the accumulation of their work activities and child care [39].

Can Table 1 be split into different tables or use a diagram to present some of this information?

Authors' response:

We followed the reviewers' suggestions. We split Table 1 into 5 different tables. We also edited tables in a better and more appropriate presentation. In addition, we created a Figure with a diagram referring to the five thematic axes of the study. In this way, the visualization of data is more fluid, efficient, and easier to understand.

Please include a "Limitations" section in the paper.

Authors' response:

We have written a section on the limitations and strengths of the study at the end of the discussion.

"The stress naturally generated in pregnancy (...)" - since stress is a concept and this was not evaluated in this study, I suggest replacing this word in the final considerations in order not to be misleading.

Authors' response:

Thank you. We replaced this word in the final considerations by “pressures”.

"The availability of information about the disease and its prevention provided more tranquility and greater adherence" - Please re-write to avoid suggesting causal effects. "Our participants referred to feel safer and also to access the services more often when provided with information," for example, would accurately express that this is a social phenomenon inferred from their interviews.

Authors' response:

Thank you very much for this comment. We rewrote the sentence according to the reviewer's suggestion.

ADDITIONAL COMMENTS FROM THE AUTHORS:

We also would like to request excluding the name of the Ph.D. "Betzabe Butron Riveros" from authorship of the study, upon her request, since she believes her contribution was not sufficient to meet the authorship criteria.

---

## [Decision Letter · Decision Letter 1]

29 Mar 2023

PONE-D-22-19394R1Experiences of women in prenatal, childbirth and postpartum care during the COVID-19 pandemic in selected cities in Brazil: the resignification of the experience of pregnancy and giving birthPLOS ONE

Dear Dr. Thomaz,

Thank you for submitting your manuscript to PLOS ONE. After careful consideration, we feel that it has merit but does not fully meet PLOS ONE’s publication criteria as it currently stands. Therefore, we invite you to submit a revised version of the manuscript that addresses the points raised during the review process.

The authors have satisfactorily met all the reviewers' requests and it is almost ready for publication. However, I would request that you respond to a final (adapted) comment from one of the reviewers:

"The authors provided valid legal and ethical reasons why excerpts of speeches of all the forty-six (46) study participants would not be made publicly available without any restriction; and also provided a non-author contact information to which a data requests may be sent. However, they must include these limitations in a Data Availability Statement in the section 'Limitations of the Study'.

We look forward to receiving your revised manuscript.

Kind regards,

Ivan Filipe de Almeida Lopes Fernandes, Ph.D.

Academic Editor

PLOS ONE

Journal Requirements:

Additional Editor Comments:

The authors have satisfactorily met all the reviewers' requests and it is almost ready for publication. However, I would request that you respond to a final (adapted) comment from one of the reviewers:

"The authors provided valid legal and ethical reasons why excerpts of speeches of all the forty-six (46) study participants would not be made publicly available without any restriction; and also provided a non-author contact information to which a data requests may be sent. However, they must include these limitations in a Data Availability Statement in the section 'Limitations of the Study'."

Reviewers' comments:

Reviewer's Responses to Questions

**Comments to the Author**

1. If the authors have adequately addressed your comments raised in a previous round of review and you feel that this manuscript is now acceptable for publication, you may indicate that here to bypass the “Comments to the Author” section, enter your conflict of interest statement in the “Confidential to Editor” section, and submit your "Accept" recommendation.

Reviewer #1: All comments have been addressed

Reviewer #2: All comments have been addressed

2. Is the manuscript technically sound, and do the data support the conclusions?

Reviewer #1: Yes

Reviewer #2: Yes

3. Has the statistical analysis been performed appropriately and rigorously? 

Reviewer #1: N/A

Reviewer #2: N/A

4. Have the authors made all data underlying the findings in their manuscript fully available?

Reviewer #1: No

Reviewer #2: No

5. Is the manuscript presented in an intelligible fashion and written in standard English?

Reviewer #1: Yes

Reviewer #2: Yes

6. Review Comments to the Author

Reviewer #1: The authors have adequately addressed all the comments I raised in my previous review of their manuscript, and thus I recommend that the manuscript has satisfied all PLOS ONE criteria for publication.

They provided satisfactory additional information on their applications for the ethical approvals from UFMA Ethics Committee and the Universidade Católica de Pelotas that authorized for the study to be carried out in the three (3) municipals study areas-São Luís, Pelotas, and Niteroi.

The authors provided valid legal and ethical reasons why excerpts of speeches of all the forty-six (46) study participants would not be made publicly available without any restriction; and also provided a non-athour contact information to which a data requests may be sent. However, they must include these limitations in their Data Availability Statement

Reviewer #2: (No Response)

7. PLOS authors have the option to publish the peer review history of their article (what does this mean?). If published, this will include your full peer review and any attached files.

Reviewer #1: **Yes: **Dr Abbas Lawal Ibrahim

Reviewer #2: **Yes: **Heloísa Garcia Claro

---

## [Author Response · Author response to Decision Letter 1]

31 Mar 2023

Thank you for allowing us to resubmit our manuscript to PLOS ONE. We are sending the revised version of the manuscript in response to this reviewer' comment. We have included this limitation in the study Discussion, as follows:

“For ethical reasons, we could not make public the database containing the transcription of all the speeches of the women interviewed, which is a limitation for open science. However, we transcribed several excerpts of statements into the manuscript, selecting representative statements from the sample, and pointing out agreements and disagreements, when present. In addition, some de-identified data may be made available upon request.”

We will be happy to make any further adjustments if necessary. 

Thank you very much.

The authors

---

## [Editor Report · Decision Letter 2]

10 Apr 2023

Experiences of women in prenatal, childbirth and postpartum care during the COVID-19 pandemic in selected cities in Brazil: the resignification of the experience of pregnancy and giving birth

PONE-D-22-19394R2

Dear Dr. Erika Barbara Abreu Fonseca Thomas

We’re pleased to inform you that your manuscript has been judged scientifically suitable for publication and will be formally accepted for publication once it meets all outstanding technical requirements.

Kind regards,

Ivan Filipe de Almeida Lopes Fernandes, Ph.D.

Academic Editor

PLOS ONE
---

## [Editor Report · Acceptance letter]

28 Apr 2023

PONE-D-22-19394R2 

Experiences of women in prenatal, childbirth, and postpartum care during the COVID-19 pandemic in selected cities in Brazil: the resignification of the experience of pregnancy and giving birth 

Dear Dr. Thomaz:

I'm pleased to inform you that your manuscript has been deemed suitable for publication in PLOS ONE. Congratulations! Your manuscript is now with our production department. 

Kind regards, 

on behalf of

Dr. Ivan Filipe de Almeida Lopes Fernandes 

Academic Editor

PLOS ONE